# An Energy-Saving Road-Lighting Control System Based on Improved YOLOv5s

Ren Tang [1], Chaoyang Zhang [1], Kai Tang [2], Xiaoyang He [1,*] and Qipeng He [3,*]

1 Research Institute of Photonics, Dalian Polytechnic University, Dalian 116039, China
2 School of Mechanical Engineering, Hefei University of Technology, Hefei 230002, China
3 Guizhou Zhifu Optical Valley Investment Management Co., Ltd., Bijie 551799, China
* Correspondence: hexy@dlpu.edu.cn (X.H.); hqpdyx@163.com (Q.H.)

**Abstract:** Road lighting is one of the largest consumers of electric energy in cities. Research into energy-saving street lighting is of great significance to city sustainable development and economies, especially given that many countries are now in a period of energy shortage. The control system is critical for energy-saving street lighting, due to its capability to directly change output power. Here, we propose a control system with high intelligence and efficiency, by incorporating improved YOLOv5s with terminal embedded devices and designing a new dimming method. The improved YOLOv5s has more balanced performance in both detection accuracy and detection speed compared to other state-of-the-art detection models, and achieved the highest cognition recall of 67.94%, precision of 81.28%, 74.53%AP$_{50}$, and frames per second (FPS) of 59 in the DAIR-V2X dataset. The proposed method achieves highly complete and intelligent dimming control based on the prediction labels of the improved YOLOv5s, and a high energy-saving efficiency was achieved during a two week-long lighting experiment. Furthermore, this system can also contribute to the construction of the Internet of Things, smart cities, and urban security. The proposed control system here offered a novel, high-performance, adaptable, and economical solution to road lighting.

**Keywords:** control system; lighting; energy saving; dimming method; improved YOLOv5s; lightweight; real-time detection





## 1. Introduction

With the development of the city, road lighting accounts for 15–19% of worldwide electricity consumption [1]. Therefore, next-generation smart street lighting requires a control system to be more intelligent and energy-saving. However, most conventional energy-saving systems are based on sensors to collect external environmental data, and sensors are very susceptible to dust and fallen leaves, resulting in low energy-saving efficiency [2]. Recently, developing artificial intelligence technology in road-lighting control systems has attracted a lot of attention, including using neural networks (e.g., [3,4]), image processing (e.g., [5,6]), and fuzzy theory (e.g., [7–9]). By continuously collecting environmental information around a light pole, they can manage the huge road-lighting network more intelligently and predict the appropriate output power for different road traffic situation. However, all of these studies have failed to maximize energy efficiency, and collecting and processing the massive and complex environment data is also a big challenge. Inspired by license plate recognition, object-detection models exhibit their capacity to accurately identify complex road conditions. This capacity, however, has not yet been accomplished in road-lighting systems. In this paper, we first introduce the improved YOLOv5s [10] into the road-lighting system, and design a matching dimming method to provide a more intelligent and energy-saving control system.

YOLOv5s is one of the most popular object-detection models, which is frequently used to detect vehicles. The prediction results of YOLOv5s will be used as the input for the

dimming method, and the dimming method determines how to change the output power. Detection ability of the model is the basis factor for the stable operation of the system. Based on this mechanism, we optimized the model in terms of detection speed and detection accuracy. Introducing the SoftPool [11] algorithm and adopting the squeeze-and-excitation (SE) [12] block to YOLOv5s improved detection accuracy, and the speed improvement came from the GhostNet [13]. With these modifications, the $AP_{50}$ increased by 4.79%, the size of the model compressed by 8.27% and the average recognition speeds per image decreased by 6.9% compared with the original YOLOv5s.

Designing the dimming method according to the model's prediction labels become the most critical factor in the energy-saving control system [14]. The design of the dimming method must first ensure the lighting safety of the road at night, while also considering its energy-saving efficiency, intelligence and usability. Then, we built a complete and intelligent dimming method for the dimming controller with four lighting modes: motor vehicle mode, non-motor vehicle mode, pedestrian mode, and none mode. Different detection results will be matched to their own lighting modes, and each of them has different priority and output power. The contributions of this study are summarized as follows:

1. For the first time, we introduced YOLOv5s into the embedded device of a road-lighting terminal.
2. We made targeted improvements to YOLOv5s, and proposed a novel, high-performance energy-saving control system.
3. We designed a complete, intelligent dimming method for the dimming controller, and the energy-saving efficiency has increased by nearly 14.1% and 35.2% compared with the same street lighting without dimming the street lighting at the same experimental site.

## 2. Related Work

### 2.1. Energy-Saving Road-Lighting Control System

Manual control, mechanical control, and computer control are three steps of the development process for a road-lighting system [15]. The research into computer control has mainly focused on the communication mode, operation and maintenance, and the management of streetlamps, e.g., [16–19]. The authors in [17] presented an energy-saving control method that can recognize the signal of body by applying infrared and sound sensors, and then dynamically control lighting output power through data processing. We believe that this sensor-based control system is insufficiently dependable, since the sensor is easily coated in dust, reducing its precision. In [7,8], the authors used fuzzy control theory to predict lighting output power. The shortcoming of fuzzy-based control systems is that they do not provide a mechanism to specify the control target. If the data are inadequate, the system's working efficiency will suffer [20]. In contrast to the above works, our method is to use image processing technology to control the lighting output power. We are not the first to do this (e.g., [5]), but what is interesting is that we introduce the improved object-detection model in road-lighting system.

### 2.2. YOLOv5s

YOLOv5 is the latest one-stage object-detection algorithm launched since YOLOv4 [21], proposed by the Ultralytics LLC team in 2020. YOLOv5 has four different architectures—YOLOv5s, YOLOv5m, YOLOv5l, YOLOv5x—and the key distinction among them is the feature-extraction modules and convolution kernels in each network's unique region. YOLOv5s is the network with the shortest network depth and feature map width, and the latter three are continuously deepening and widening on this basis [22]. YOLOv5s' most important feature is its superior flexibility, which makes it easy to deploy quickly on the embedded device side, which is one of the primary reasons we chose YOLOv5s. The network structure of YOLOv5s is shown in Figure 1.

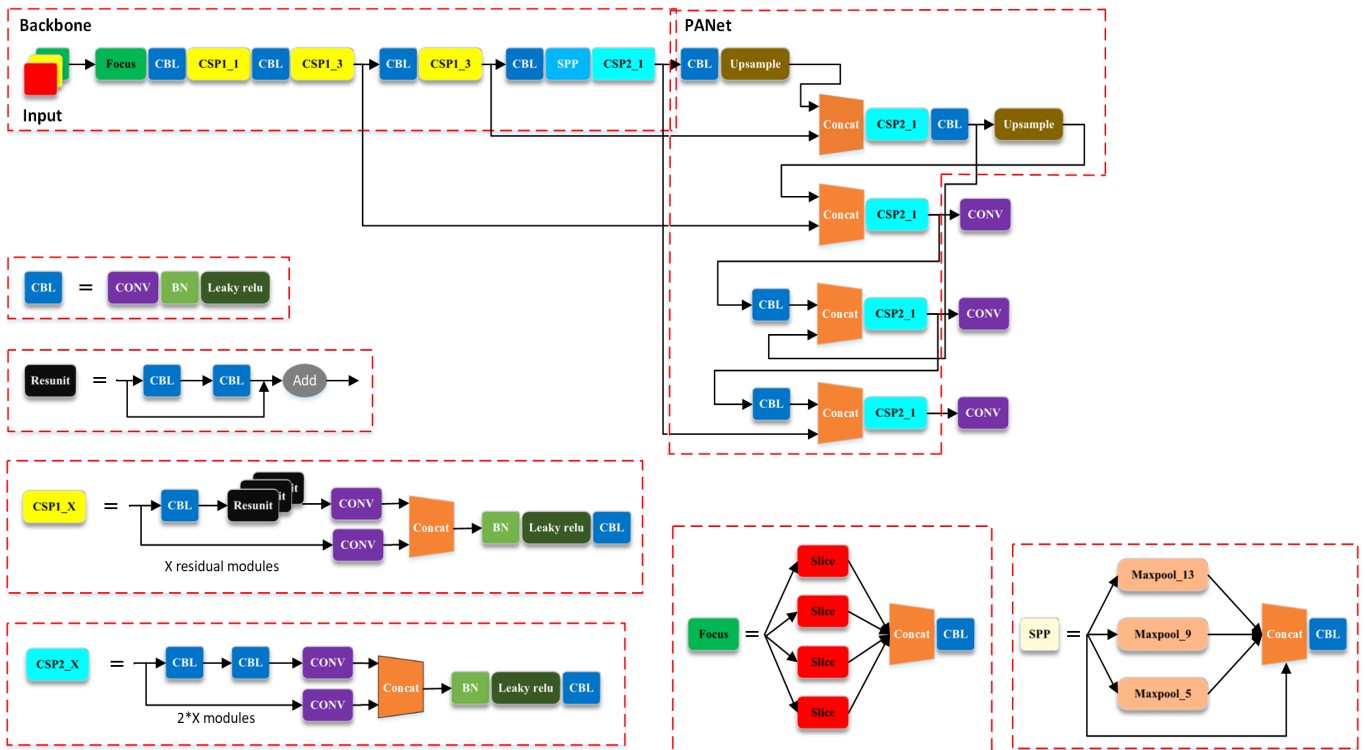

**Figure 1.** The network structure of YOLOv5s.

*2.3. YOLOv5s-Based System*

YOLOv5s is widely used in traffic detection, and many studies have shown its ability to be incorporated with embedded devices and detect in real time.The authors in [23] proposed a large-scale fusion module based on YOLOv5s' backbone network to detect vehicles on the road. In [24], the authors deployed the improved YOLOv5s on embedded devices to achieve a low-cost and real-time vehicle-exhaust detection. In [25], the authors proposed an automatic license-plate recognition technique based on deep learning and trained YOLOv5 to detect license plates in traffic videos. These YOLOv5s-based systems provided a lot of inspiration, and made us think of introducing YOLOv5s into road lighting.

## 3. Methodology

*3.1. Proposed System*

The reported control system is composed of a high-definition camera, an object-detection chip, a control chip, a dimming controller, and an LED lighting board. The high-definition camera is installed on the top of the pole, with a 2.8 mm focal length and a resolution of 1920 × 1080. We deployed YOLOv5s in the embedded chip, which drastically reduced the system response time. The dimming controller included four gears, which matched to the four lighting modes that we designed. Figure 2 depicts how our control system works: the high-definition camera transmits the captured video frames to the object-detection chip, which will report the prediction labels to the control chip after completing the image processing. Then, the prediction labels will be used as the input of the dimming method for a series of data-processing stages in the control chip. Lastly, the dimmer controller will activate a lighting mode on the LED lighting board according to the output of the dimming method.

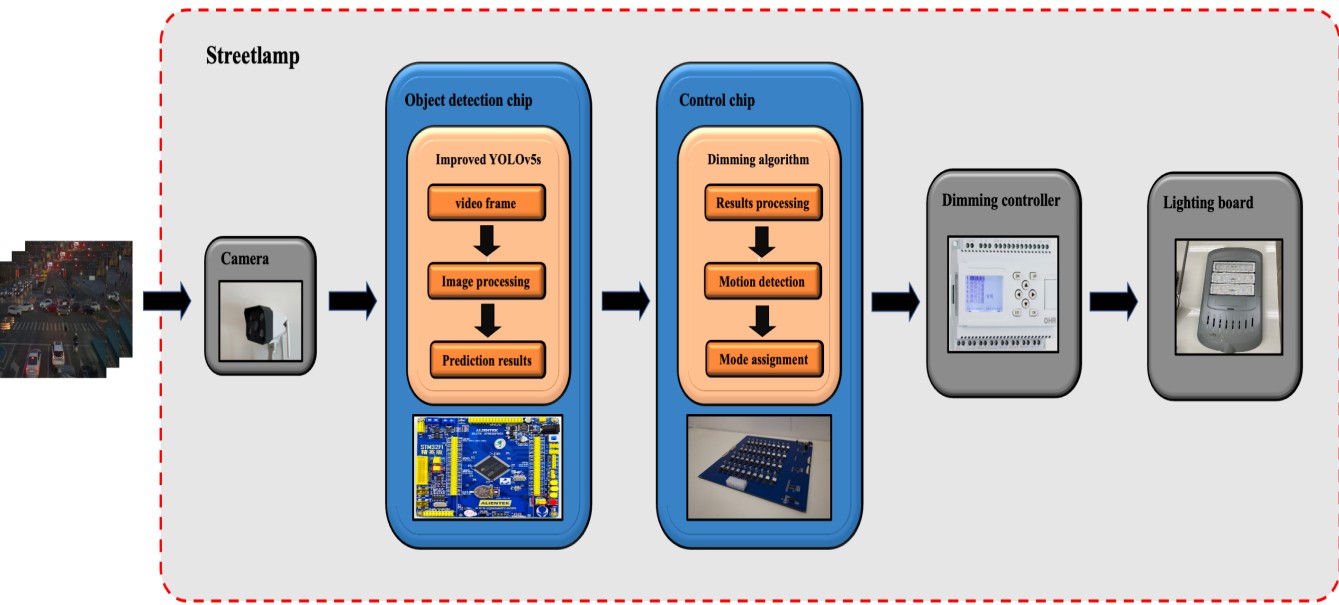

**Figure 2.** The structure of the proposed system.

### 3.2. Dimming Method

The proposed method achieved highly complete dimming control by involving three important modules: prediction result processing (PRP), motion-detection algorithm (MDA), and mode assignment (MA). PRP is a parameter-extraction module, which reports the required motion-detection parameters to the database and MA module by analyzing and processing the model prediction results. Three significant types of data, including the classification of predicted labels, the number of each label, and the coordinates of bounding boxes in the prediction results, will be extracted and saved in the database as background data. Simultaneously, the same data from the current video frame will be submitted to the MDA module as incoming data, i.e., the PRP module provides an input for the MA module with three-dimensional data of classification, quantity, and coordinates, which is also necessary for motion detection.

The three basic types of motion-detection methods are background subtraction, temporal differencing, and optical flow. Background subtraction is the most popular motion-detection method and consists of the differentiation of moving objects from a maintained and updated background model, which can be further grouped into parametric type and non-parametric type [26]. The proposed MDA module is a simpler motion-detection algorithm that combines background subtraction and YOLOv5s prediction results. The prior background frame $B_{t-1}(x, y)$ and the incoming frame $I_t(x, y)$ are then combined with the current background image. The following basic adaptive filter is used to create the background model:

$$B_t(x, y) = (1 - \beta)B_{t-1}(x, y) + \beta I_t(x, y) \tag{1}$$

$\beta$ is an empirically adjustable parameter. The binary motion-detection mask $D(x, y)$ is defined as follows:

$$D(x, y) = \begin{cases} 1, & \text{if } |I_t(x, y) - B_t(x, y)| > \tau \\ 0, & \text{if } |I_t(x, y) - B_t(x, y)| \leq \tau \end{cases} \tag{2}$$

where $\tau$ is the preset pixel threshold, and the pixel blocks exceeding this threshold in the background subtraction process will be regarded as motion labels and report to the MA module.

Finally, after counting the classification and quantity of the motion labels, the MA module will determine the priority of each label's lighting mode, which is also the final output of the control chip. Figure 3 is the flow chart of the proposed dimming method.

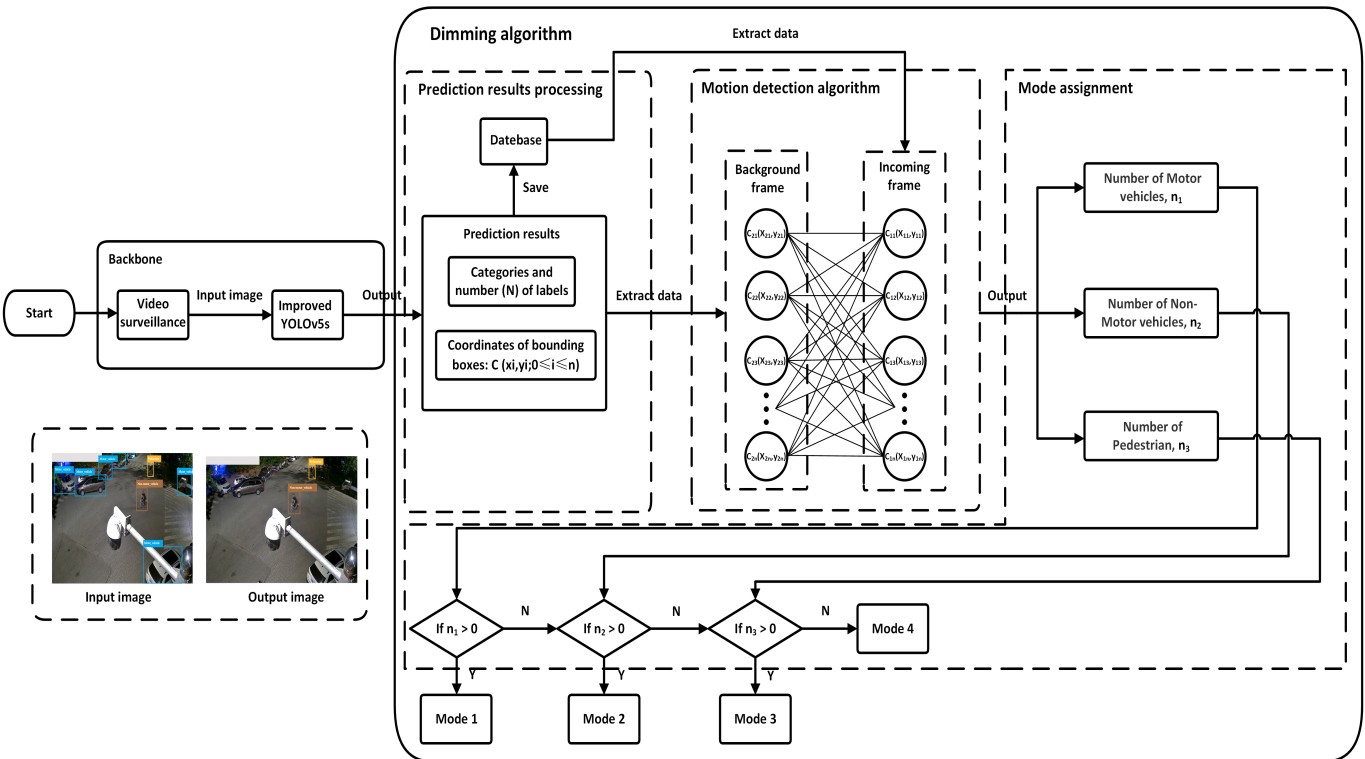

**Figure 3.** The proposed dimming method.

Road-lighting design standards can provide people with safe, reliable, and compliant night lighting. There are many road-lighting standards that can be referred to at present, and the most widely used ones are the Road-Lighting Standard Recommendation for the Lighting Roads for Motor and pedestrian Traffic promulgated by the International Commission on Illumination (CIE) in 2020, the road-lighting standard IESNA-RP-8-00, and China's urban road-lighting standard CJJ 45-2006 [27]. The experimental site of this study is in China, so the standard we used is CJJ 45-2006. Average road surface luminance, average road surface illuminance, overall uniformity of road surface luminance, uniformity of road surface illuminance and glare threshold increment are five important values of road lighting. The motor vehicle and non-motor vehicle traffic (including pedestrians) road-lighting standard values in CJJ 45-2006 are listed in Tables 1 and 2, respectively.

**Table 1.** Motor vehicle traffic road-lighting standard values.

| Road type | Average Surface Luminance | Average Surface Illuminance | Overall Uniformity of Luminance | Uniformity of Illuminance | Glare Threshold increment |
|---|---|---|---|---|---|
| Major road | 1.5/2.0 | 20/30 | 0.4 | 0.4 | 10 |
| Local road | 0.75/1.0 | 10/15 | 0.4 | 0.35 | 10 |
| Conflict areas of major road and conflict road | / | 30/50 | / | 0.4 | / |

**Table 2.** Non-motor vehicle traffic (including pedestrians) road-lighting standard values in residential area.

| Night Traffic Flow | Average Surface Illuminance | Minimum Light Surface Illuminance |
|---|---|---|
| High | 10 | 7.5 |
| Middle | 7.5 | 5 |
| Low | 5 | 1 |

The factory settings of the street lighting with LED light sources, which we used in line with the above lighting standard, i.e., when the lights are installed at a suitable height, can adjust its output power to make the above five values meet the requirements of CJJ 45-2006. The dimming controller contains a chip that can adjust the output power by changing the voltage, which is why our system can save energy. The motor vehicles have the highest requirements for night lighting, which is because drivers need to observe obstacles through the window, and they are all moving at a high speed [28]. Considering that non-motor vehicles move faster than pedestrians, they need to observe obstacles earlier, and there is low or no traffic flow on the road in the middle of the night. According to Tables 1 and 2, we designed four lighting modes—motor vehicle mode, non-motor vehicle mode, pedestrian mode, and none mode—which have different output powers and dimming priorities. Motor vehicles have the highest dimming priority, followed by non-motor vehicles, and pedestrians, i.e., motor vehicle mode turns on whenever a motor vehicle is detected, regardless of other types of traffic on the road. The output power of the motor vehicle mode is 120 W, non-motor vehicle mode, pedestrian mode and none mode are 100 W, 70 W and 50 W, respectively. Figure 4 shows the dimming circuit.

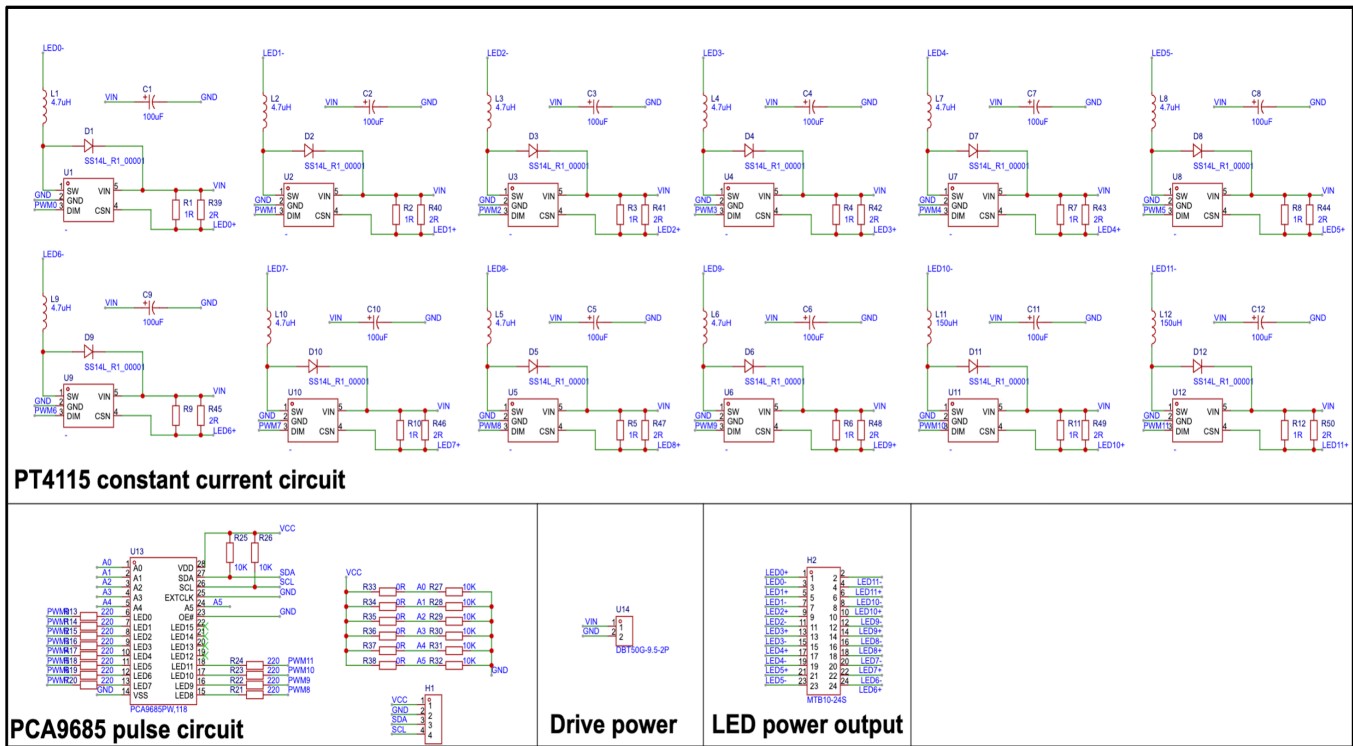

**Figure 4.** The dimming circuit.

### 3.3. The Improved YOLOv5 Network Framework

We made three specific improvements to YOLOv5s to improve its traffic-detection performance during nighttime. First, the improvements to detection speed allow the control system to dimming quickly, which is also the most important feature of a real-time detection system. GhostNet is a lightweight convolutional network proposed at CVPR2020 by Huawei's Noah Lab. Through a series of linear transformations at cheap cost, GhostNet can generate many ghost feature maps to retain interaction information [29]. The core idea is to divide the original convolution operation into two stages. The first stage performs a small amount of convolution calculations, and the second stage performs block-by-block linear convolution on the basis of the feature map obtained in the first stage to generate ghost feature maps, and finally combines them to obtain a large number of feature maps. Figure 5 shows how the ghost module generates feature maps.

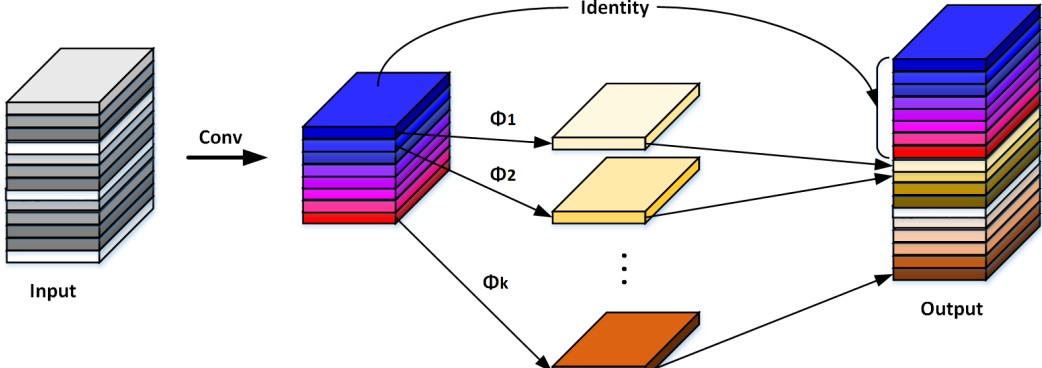

**Figure 5.** The ghost module.

For input data $X \in R^{c \times hw}$, $c$, $h$, and $w$ are the input channel numbers, height, and width of the feature map, respectively. After one convolution, it is $n * h' * w'$, the size of the convolution kernel is $k$, and the size of the convolution kernel of a linear transformation is $d$ after $s$ transformations. We can conclude from this that the theoretical speed-up ratio of the ghost module is:

$$r_s = \frac{n \cdot h' \cdot w' \cdot c \cdot k \cdot k}{\frac{n}{s} \cdot h' \cdot w' \cdot c \cdot k \cdot k + (s-1) \cdot \frac{n}{s} \cdot h' \cdot w' \cdot d \cdot d} = \frac{c \cdot k \cdot k}{\frac{1}{s} \cdot c \cdot k \cdot k + \frac{s-1}{s} \cdot d \cdot d} \approx \frac{s \cdot c}{s+c-1} \approx s \quad (3)$$

$n/s$ is the number of output channels in the first transformation; the identity map does not need to be computed, but it is also counted as a part of the second transformation, which is the reason the ghost module can accelerate the inference process.

Softpool is a fast and efficient method for exponentially weighted activation downsampling, which can retain more information in the reduced-activation refined downsampling to improve classification accuracy [11]. Most pooling operations rely on different combinations of max pooling and average pooling, while SoftPool's work is based on a the SoftMax weighting operation to preserve the input's basic properties [30]. We introduce SoftPool into the SPP module to optimize the original pooling operation, which improves the detection accuracy under the same computing load and memory conditions. The downsampling activation mapping process of SoftPool is shown in Figure 6.

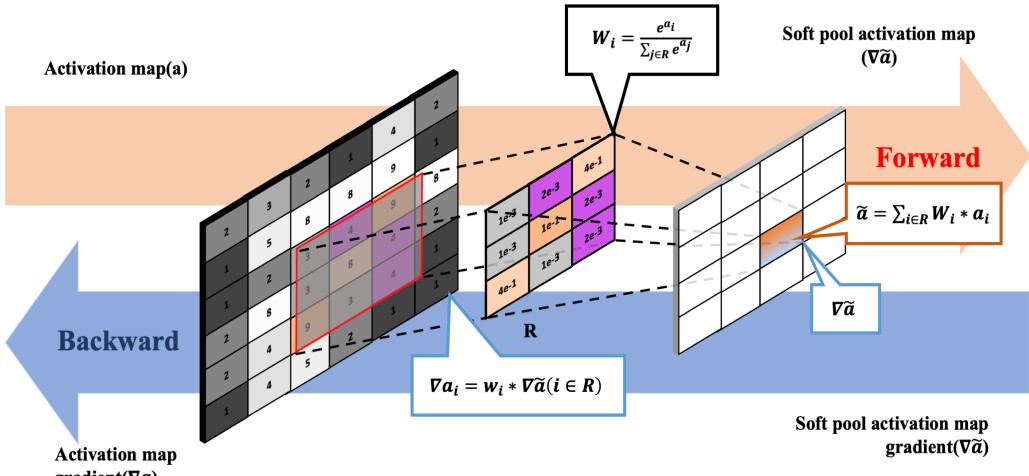

**Figure 6.** SoftPool calculation.

The main concept is the use of SoftMax to nonlinearly calculate the feature weight value of region R according to the feature value $W_i$:

$$W_i = \frac{e^{a_i}}{\sum_{j \in R} e^{a_j}} \tag{4}$$

The weight $W_i$ can ensure the transmission of important features, and the eigenvalues in the region $R$ will have at least a preset minimum gradient during reverse transfer. After obtaining the weight $W_i$, the output is obtained by the eigenvalues in the weighted region $R$:

$$\widetilde{a} = \sum_{i \in R} W_i * a_i \tag{5}$$

Lastly, we introduce the SE block for our model, which can improve the performance of detection models with minimal computational cost. SENets, which are stacked by SE modules, can extract relevant features by combining spatial and channel-wise information inside local receptive fields [12]. The SE block is a typical attention module that can handle the loss problem produced by the varying relevance of different channels of the feature map during the convolution pooling process. "Squeeze" is the first operation of the SE block, which uses global average pooling to create statistics for each channel to achieve the purpose of squeezing global spatial information into a channel descriptor [12]. $F_{tr}$ stands for any given information: $X \to U$, $X \in R^{H' \times W' \times C'}$, $U \in R^{H \times W \times C}$, and the way to obtain a statistic $z \in R^c$ is to shrink $U$ by a pooling size $H \times w$, and the $z's$ c-th element is computed as follows:

$$z_c = F_{sq}(U_c) = \frac{1}{H \times w} \sum_{i=1}^{H} \sum_{j=1}^{W} u_c(i, j) \tag{6}$$

The second operation of the SE block is "Excitation", after the squeeze operation, to fully capture channel-wise dependencies, using a fully connected neural network to perform a nonlinear transformation on the result. However, two requirements must be satisfied to carry out the excitation operation:

1. Nonlinear interactions between channels must be figured out.
2. Each channel should be assured to have a matching output, and instead of a one-hot vector, a soft label can be generated.

To meet these requirements, using a simple gating method for sigmoid activation, and rescaling the transformation output $U$ with the activation yields the block's final output:

$$\widetilde{X_c} = F_{scale}(U_c, S_c) = S_c \cdot U_c \tag{7}$$

$F_{scale}(U_c, S_c)$ refers to channel-wise multiplication between the feature map $U_c \in R^{H \times W}$ and the scalar $S_c$, and $\widetilde{X} = \left[ \widetilde{X_1}, \widetilde{X_2}, \cdots, \widetilde{X_c} \right]$. The structure of a squeeze-and-excitation block is shown in Figure 7.

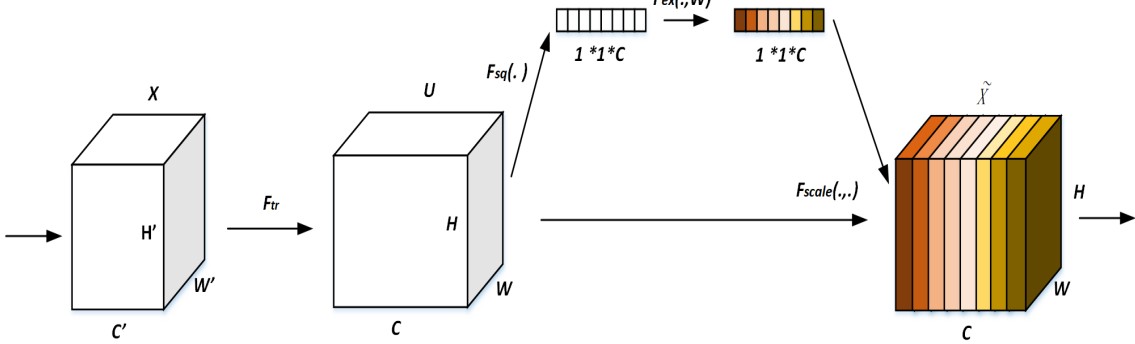

**Figure 7.** A squeeze-and-excitation block.

With these three adjustments, the improved YOLOv5s network outperforms the original network in terms of detection speed and accuracy, and the network structure is shown in Figure 8.

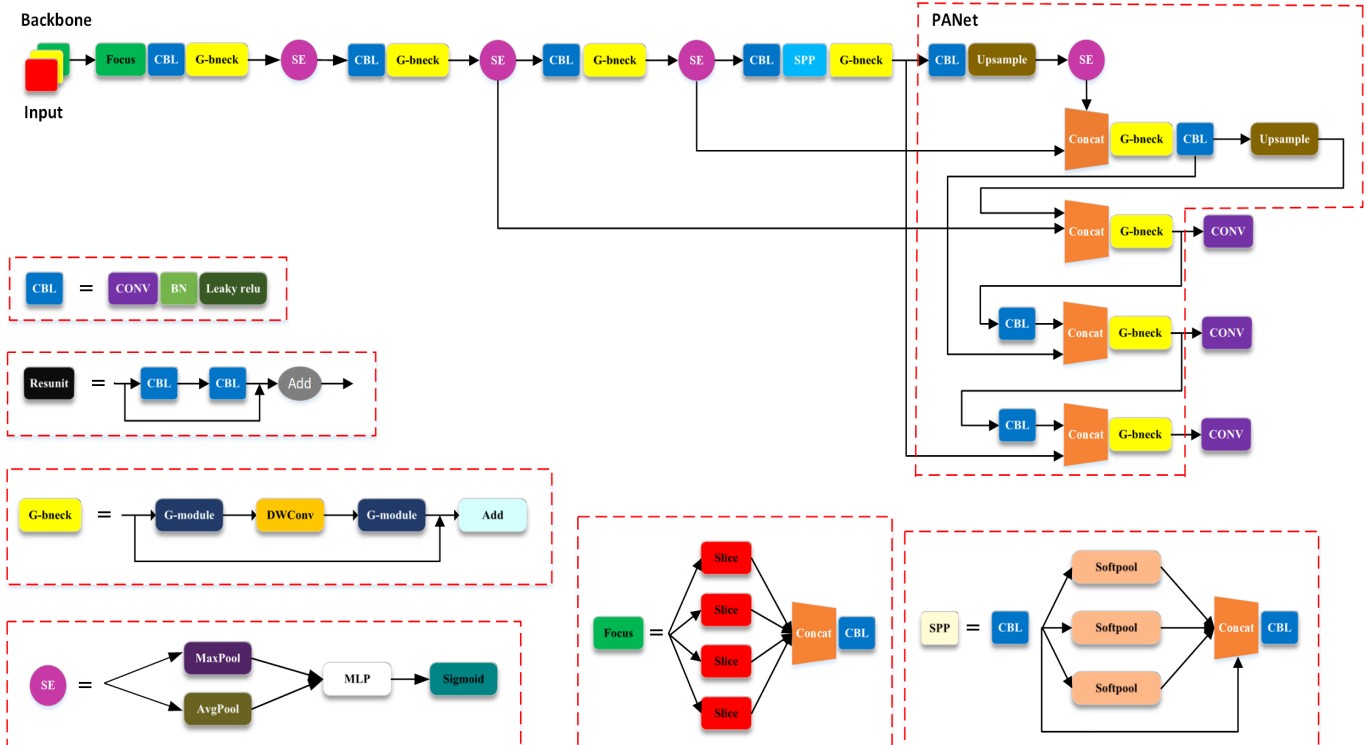

**Figure 8.** The network structure of improved YOLOv5s.

### 3.4. The Traffic Dataset

The dataset we used is DAIR-V2X [31], which is the world's first vehicle-infrastructure cooperative 3D object-detection dataset released by the Institute of Artificial Intelligence, Tsinghua University, in February 2022. The single-infrastructure-side dataset contains 10,084 traffic images of the Beijing High-level Autonomous Driving Demonstration Zone with a resolution of 640 × 640. This dataset includes a variety of weather conditions at night such as clear, foggy, and rainy. Moreover, this dataset's data are complete, including desensitized original image and point cloud data, annotation data, timestamp, calibration file, etc. We categorize the traffic annotations into three groups: motor vehicle, non-motor vehicle, and pedestrian. Instead of using the entire content of DAIR-V2X, the images we chose are from each period of the night scene. There are 6348 traffic images in the reworked training set, which still has a very high object density, with 125,654 (80,047 + 27,406 + 18,201) labeled objects, and covers almost all kinds of vehicles. Figure 9 depicts a sample image of the dataset.

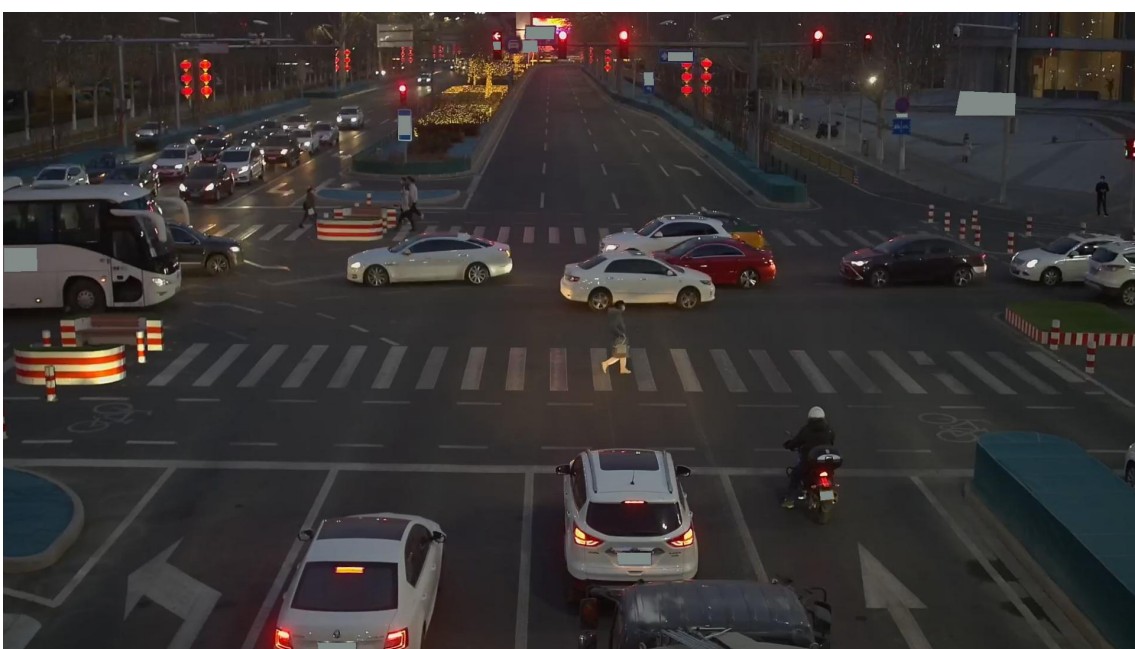

**Figure 9.** A dataset sample image.

### 3.5. Experimental Environment

The experimental site of this study is the intersection of Yushi Street and Yushui Street, Ganjingzi District, Dalian City, Liaoning Province, surrounded by residential areas, parks and universities. In CJJ 45-2006, the type of road where the experimental site is located is a "conflict area of major road and conflict road", which is defined as a high traffic flow area. Therefore, we adjusted the output power of motor mode, non-motor mode and pedestrian mode to meet the requirements of the standard. We collected 1058 images of this location as a test set, with a ratio of 6:2:2. The whole dataset was divided into training (6348 images), validation (1058 images), and testing (1058 images). The training environment is Pytorch and the GPUs we used are two Nvidia GTX 3090, each with 24 GB VRAM. In the test experiment, we just used a single Nvidia GTX 3090 GPU and an i7-9700 CPU working at 3.00 GHz. The momentum, batch size, decay of weight, training epochs, and learning rate were set to 0.937, 48, 0.0005, 300, and 0.01, respectively.

### 3.6. Evaluation Metrics

Precision and recall are the most used evaluation metrics in the field of object detection [32]. The proportion of accurate results predicted by the model to "all predicted results" is known as the precision value, and the recall value indicates the proportion of the accurate results predicted by the model to "all positive samples". We use (8) and (9) to calculate precision and recall values. In practice, it is difficult for us to make the value of these evaluation metrics high. Generally speaking, the higher the precision value, the lower the recall value.

$$precision = \frac{TP}{TP + FP} \tag{8}$$

$$recall = \frac{TP}{TP + FN} \tag{9}$$

The number of objects that were found in the dataset is shown by TP (true positives). The number of objects identified incorrectly by the detection model is shown by FP (false positives). The number of objects missed by the detection model is represented by FN (false negatives). When the IOU threshold [32] is set to 0.5, for n samples of a certain category, if it has m positive examples, each positive example corresponds to a recall rate *R* value

$(1/m, 2/m, \ldots, 1)$, calculates the maximum precision P for each recall rate, and then the AP is calculated as follows:

$$AP = \frac{1}{m}\sum_{i}^{m} P_i = \frac{1}{m}*P_1 + \frac{1}{m}*P_2 + \cdots \frac{1}{m}*P_m = \int_0^1 P_R D d_R \tag{10}$$

The AP is for a specific class, and a dataset typically has several categories, and the mAP is calculated by averaging the AP values of all classes in the dataset:

$$mAP = \frac{1}{C}\sum_{j}^{C} AP_j \tag{11}$$

Therefore, in mAP, P represents the maximum accuracy of a sample, AP represents the average accuracy of a class of samples, and mAP is the average accuracy of the data set.

## 4. Results and Discussion

### 4.1. Detection Performance

Our experiments compared the proposed algorithm with the current state-of-the-art detection algorithms including YOLOv5s, YOLOv4, YOLOv3, and SSD. In the proposed system, both detection accuracy and detection speed significantly impact experimental results. The detection accuracy is the guarantee of road lighting, and the detection speed has a direct impact on the system's dimming reaction time. After setting the IOU threshold to 0.5, the improved YOLOv5s achieves a better balance between detection accuracy and speed in the comparative experiments. To this end, we trained 200 epochs on the DAIR-V2X dataset, and in Figure 10, we exhibit the mAP of each model (ours, YOLOv5s, YOLOv4, YOLOv3, SSD) as a function of training epochs. The variation of mAP with training epochs can be evaluated based on comprehensive precision and recall values, and the graph can also provide a lot of information about the model performance.

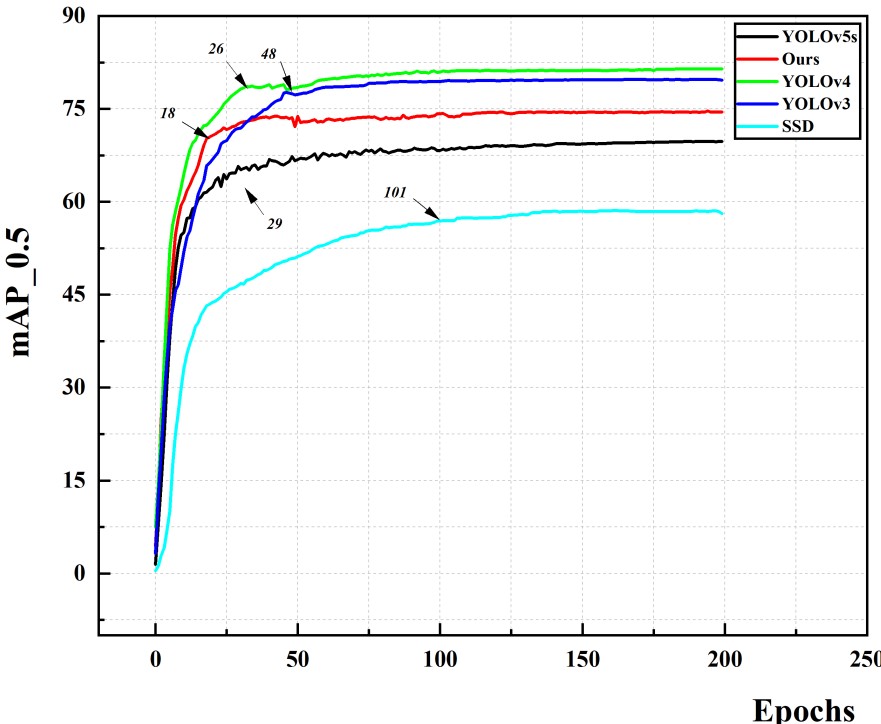

**Figure 10.** The mAP map of each model evolves with the training epoch.

In our test, YOLOv4 has the highest AP$_{50}$ value (0.8149), followed by YOLOv3 (0.7799), the proposed model (0.7453), YOLOv5s (0.6974), and SSD (0.5853). In comparison with

YOLOv5s, our mAP value is 4.79 percentage points higher, which is a significant improvement. Furthermore, our curve showed a convergence trend in the comparison experiment. After the 29th epoch, YOLOv5s began to converge, while the SSD curve remained flat, and even progressively converged at the 101st epoch, which also demonstrates that our model has a stronger learning ability. This also shows that the improved SPP module and the introduced attention mechanism can improve the model's detection ability at night. We contrasted the replacement of the ghost module, the upgraded SPP module, and the addition of the attention mechanism to verify our hypothesis, and the detection results are plotted in Figure 11.

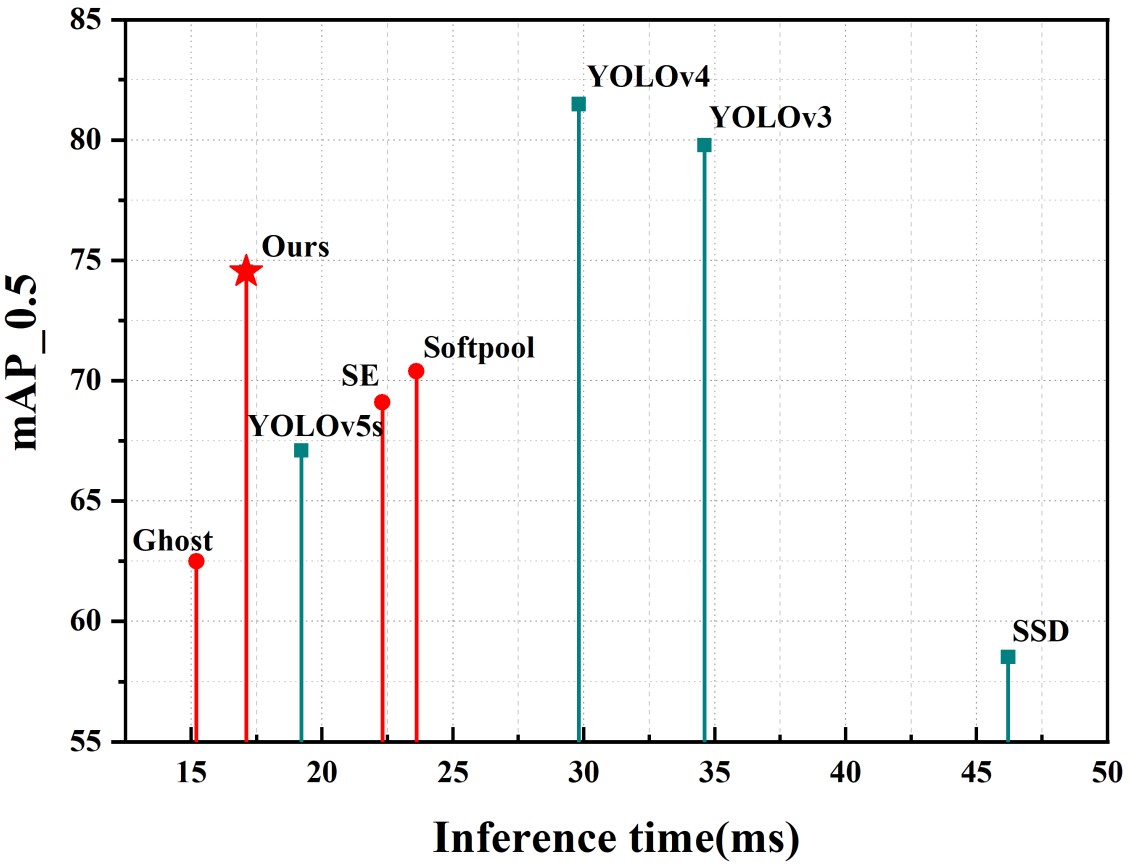

**Figure 11.** The $AP_{50}$ and inference time of each model.

The introduction of SE block and the improved SPP module improved the mAP value of the model to varying degrees (1.99% and 3.29%, respectively). In [11,12], the mAP value improvement was more noticeable. Although the improvement in mAP is minor, it comes at the cost of increasing inference time (1.99 ms and 3.29 ms, respectively). The ghost module can be used to supplement the above modules, which is why our model surpasses YOLOv5s in terms of inference time and mAP value. Since our system needs to deal with complex road traffic at night and offer people with safe and reliable night lighting, we require the object-detection model to include detection accuracy and speed. From the experimental results, the proposed system performs very well in these two aspects. It inherits YOLOv5s' amazing detection speed while also improving detection accuracy (74.53% and 17.1 ms, respectively). We saved all the label detection maps from the test experiments, in which we found a set of label maps (Figure 12) that can identify the detection capabilities of our model.

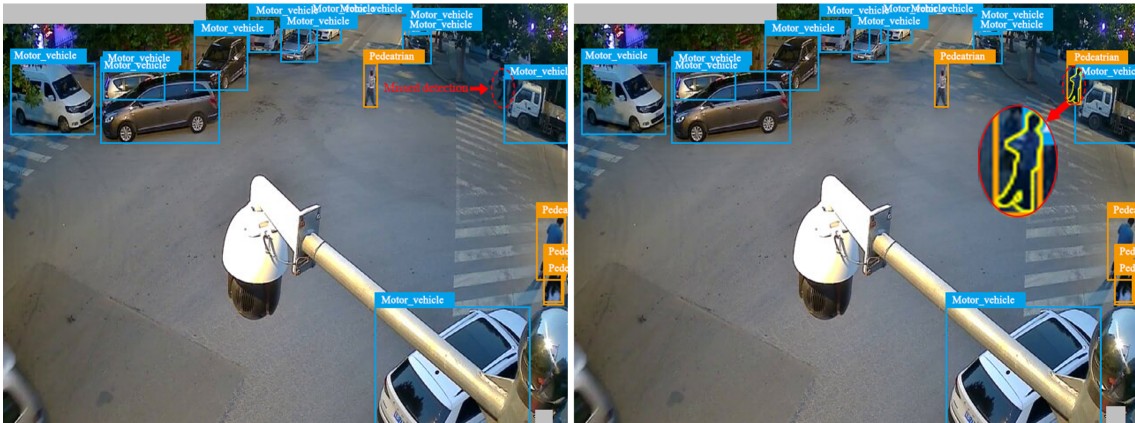

**Figure 12.** A set of label images obtained by detecting the test set with YOLOv5s and the suggested model, respectively. For the same image, YOLOv5s fails to detect the pedestrian in the shadow (small objection), but our proposed model correctly assigns the label to the objection (pedestrian).

### 4.2. Energy-Saving Experiment

Based on the surveillance video we took at the experimental site, we conducted lighting experiments in the laboratory. Before the experiment started, we did a simple verification of whether the four lighting modes met the CJJ 45-2006: The height of the light pole at the experimental site is about 7 m and the height of the street lighting from the ground is 6.5 m. When the output power of the LED lighting board is 120 W, the light environment of the road surface meets the requirement of Table 1's conflict areas of major road and conflict road. Therefore, we set the output power of the motor vehicle mode at 120 W. Similarly, the light environment of the pedestrian mode should meet the high traffic flow's requirements in Table 2. Similarly, the light environment of the pedestrian mode should meet the high flow requirements in Table 2, and the output power is 70 W. None mode (50 W) corresponds to the low traffic flow in Table 2. Since the speed of non-motorized vehicles is also very fast, especially electric bicycles, they need to find obstacles in the road earlier than pedestrians, so we set the output power for the non-motor mode to be 100 W.

#### 4.2.1. Traffic Flow

We collected images for a total of 14 days (7:12 p.m. to 6:30 a.m., 25 June to 8 July 2021), and Figure 13 shows the traffic situation throughout the night at the intersection of Yushi and Yushui on 28 June 2021. There were very few labels recorded between 12 p.m. and 4 a.m., when we count labels at 5 min intervals; indeed, no label was recorded at all in many time periods. Therefore, the energy-saving efficiency of the proposed system at this experimental site is very impressive. Traffic flow statistics are listed in Table 3.

**Table 3.** Traffic flow statistics.

| Time | Motor Vehicle | Non-Motor Vehicle | Pedestrian |
|---|---|---|---|
| 7.15–9.00 | 476 | 286 | 926 |
| 9.00–11.00 | 263 | 92 | 456 |
| 11.00–01.00 | 64 | 34 | 49 |
| 01.00–03.00 | 27 | 9 | 49 |
| 03.00–05.00 | 52 | 5 | 62 |
| 05.00–07.00 | 348 | 24 | 265 |
| total | 1516 | 450 | 1765 |

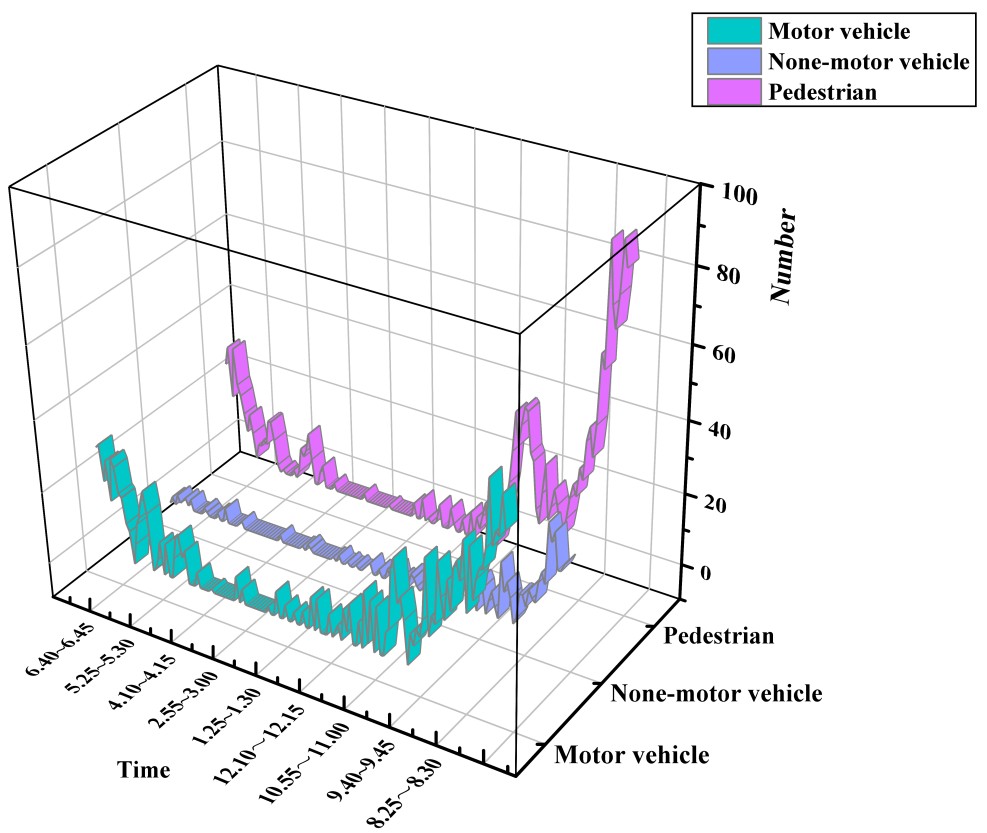

**Figure 13.** Traffic flow graph.

#### 4.2.2. Energy-Saving Efficiency

The street lighting at the experimental site was turned on from 7:00 p.m. to 6:00 a.m. the next day, and we saved the surveillance videos from 25 June to 8 July 2021, for a total of 14 days. Our energy-saving experiment is based on a surveillance video, which will provide images for the improved YOLOv5s in the object-detection chip. We recorded the power consumption of the proposed system for these 14 days, the power consumption of the original street lighting at the experimental site during the same period of time, and calculated the power consumption of street lighting with an LED light source that cannot adjust its output power according to the road conditions (output power is 120 W). Figure 14 is a comparison of power consumptions.

Label 1 in Figure 14 represents the power consumption of the original street lighting at the experimental site, label 2 represents the power consumption of street lighting with an LED light source that cannot adjust its output power according to road conditions. Since we use the electricity consumed by the street lighting to represent its energy consumption, some external factors will cause fluctuations. These external factors are very complicated, such as the heat dissipation of street lights. High-temperature weather will keep the lighting at a relatively high temperature, resulting in slightly higher power consumption. Similarly, rainfall and windy weather will also affect the power consumption of street lights. Lastly, we tabulate the average power consumption and energy-saving efficiency (compared to the proposed system) of these three lights in Table 4.

**Table 4.** Power consumption statistics.

| Lighting | Power Consumption (kwh) | Energy-Saving Efficiency |
|----------|-------------------------|--------------------------|
| 1 | 1.59 | 135.2% |
| 2 | 1.2 | 114.1% |
| Ours | 1.03 | 1 |

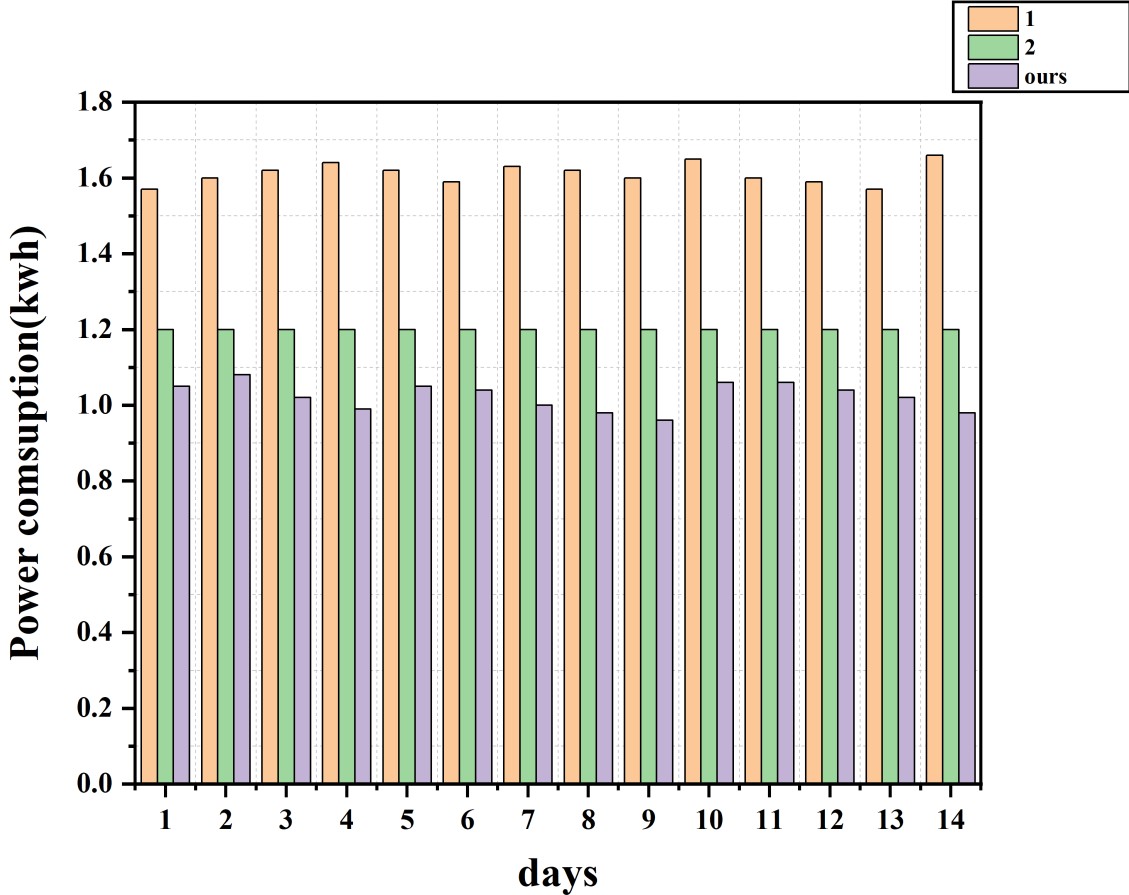

**Figure 14.** Comparison of power consumption of three lights.

In conclusion, the proposed control system has high energy-saving efficiency, which is 35.2% lower than the power consumption of street lighting at the experimental site and 14.1% lower than the lamp with an LED light source that cannot adjust its output power according to the road conditions.

In fact, the energy consumption of our proposed system is slightly higher than the experimentally obtained value, because we only record the energy consumption of the lights and the total energy consumption includes the energy consumed by processing data, communicating, driving cameras, etc. Because they consume very little energy, it is difficult to measure their value in a single street-lighting system. However, in a huge road-lighting system, their energy consumption may be very considerable. This is also the research work we will do next: a district-level energy-saving road-lighting system solution for "smart cities" based on energy consumption, cost, and construction difficulty of each part.

## 5. Conclusions

In this paper, we successfully applied the improved YOLOv5s in the field of road lighting, and designed a complete and intelligent dimming method. Our improvements to YOLOv5s are mainly to balance detection speed and detection accuracy in complex road conditions, and we believe that the improved network can also be applied to more scenarios in the future. Furthermore, we plan to promote and apply our system in other complex traffic areas, such as urban business districts, large communities, schools, etc. To do this, we need to ensure the efficient work of a single street lighting and the sharing of information among street lights. We believe that the upgraded street lights can be applied in more road scenarios and make more contributions to the sustainable development of the city and the construction of the "smart city".

**Author Contributions:** Investigation, R.T.; Methodology, R.T., C.Z., K.T., X.H. and Q.H.; Software, C.Z. and K.T.; Validation, R.T.; Visualization, R.T.; Writing—original draft, R.T.; Writing—review and editing, R.T., K.T., X.H. and Q.H. All authors have read and agreed to the published version of the manuscript.

**Funding:** This research was funded by Guizhou Zhifu Optical Valley Investment Management Co., Ltd. and APC was funded by [Bi Jie He Zi]: 2022,02.

**Institutional Review Board Statement:** Not applicable.

**Informed Consent Statement:** Not applicable.

**Data Availability Statement:** Not applicable.

**Acknowledgments:** We would like to thank the Dalian Streetlamp Management Office, which is affiliated to the Dalian Municipal Government, for providing us with light poles and experimental sites. We would like to thank the technical and experimental equipment support from Guizhou Zhifu Optical Valley Investment Management Co., Ltd. We sincerely thank the reviewers for their critical comments and suggestions for improving the manuscript.

**Conflicts of Interest:** The authors declare that they have no known competing financial interests or personal relationships that could have appeared to influence the work reported in this paper.

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
