# Peer review of "An Energy-Saving Road-Lighting Control System Based on Improved YOLOv5s"

_computation, doi:10.3390/computation11030066_

Round 1
Reviewer 1 Report
Nice job, and i do see the Camera application options in streets or even lighting control. Research is valuable, but it is more in the field of IT domain. Some comments, that could be improved:
1) Line 146-147. What determines such parameters? There are specific road standards that define needed parameters. Cd and lux are just one part of the coin, and Dialux or other simulation software determines at which power (W) such light quality will be obtained... and these values are different across streets.... So this approach is very vague and blurry, or I would say not applicable unless you have a certain method that has been practically prooved, but not referenced in this article. The dimming algorithm must be renamed, as it is not real dimming, as it is related to either LED power, current (A) or at least diming % (that can be then extrapolated to real control signal). How did you do that in the experiments you mention? Give more explanation and electrical circuit connections.
2) How do you estimate "energy efficiency", and "energy savings", no formulas were given. Please revise and add them. Fig.13. is really unclear... and hard to read... don't use this format to compare different tech energy consumption. How did you measure or calculated it? What is this tradition you refer? Give more details. I doubt that you can get such savings % unless traditional control is indeed very badly designed.
3) Fig.9. is good, but I would add one more graph nearby with a total count of traffic users. Is it verified with another sensor or someone did count manually through video? How precise is this model?
4) Surely References can be improved, not clear the reference to 30, 33-39, how they are related to your research. Do not reference "word", make a reference if you compare or use the method or approach described there.
Reviewer 2 Report
My area of expertise is lighting technology, so I will only comment on parts of the article related to road lighting and general observations. I will not go into the correctness of the description and operation of the algorithm.
First general remarks. Most of the terms in the article related to lighting technology are incorrect. For example, instead of "Streetlamp" the term "Road lighting luminaire" should be used. Further, lux (lx) is the unit for illuminance and not illumination as the cd/m2 is the unit for luminance and not brightness and 2 after m should be superscript as it means "square meter". Check all "lighting terms" and correct them if needed.
On most of the figures, the text is too small to be readable. What is shown in Figure 11? If it is a value of mAP_0.5 for different interference times and individual protocols, why are the points connected with lines? What do these lines show? Is it possible to choose something that is between YOLOv3 and SSD and has a certain value of mAP_0.5 at an interference time of 40ms? Similarly, what is shown in Figure 13? If this is the energy, as the y-axis describes, that the lighting consumes in one day, why are the daily values connected by lines? For example, what does the black line value between day 1 and day 2 represent? But if this is the value of the energy (power integral) that the lighting consumes from the time it is turned on at the start of day 1, why aren't the functions monotonically increasing? A falling function in this case means that the lighting is generating energy???
It is also not clear what is meant by "energy consumption of common street lamps, time-controlled street lamps, and light-controlled street lamps". If "common street lamps" (correct would be "common street lighting luminaires") are luminaires without luminous flux control of any kind, then their energy consumption should be the same every day. Why then in Figure 13 their energy consumption change from day to day? The same is valid for the "time-controlled street lamps". The energy can only slightly change in the case of "light-controlled street lamps" where it could increase or decrease slightly, depending on whether the length of the day is longer (spring) or shorter (autumn).
Further, the factors in the equations are not properly explained (below the equations), so the derivation cannot be followed.
Remarks about lighting. Stating the luminance levels (lines 146, 147) is not appropriate because, for lighting the intersection, the design should consider needed illuminances and not luminances. In addition, in such cases, it is also not possible to unambiguously determine the connection between illuminance and luminance, because it can be different in different directions of view. Also, the assumed control strategy of the luminous flux of luminaires is not the right one. It is not advisable to change illuminance levels in steps as this might disturb drivers and cause accidents. So soft change should be incorporated and this could influence savings.
Since there are many other things in the article that need to be corrected, I suggest that the authors expand their knowledge in the field of lighting technology, check what graphs are suitable for displaying certain data and how to describe the factors of the equation... and then thoroughly rework the article.
Round 2
Reviewer 1 Report
Table 4, must be revised: "Energy saving efficiency" is wrongly calculated, as you now have some "energy consumption difference" not energy efficiency. The correct way is to compare to existing tech, respectively "ours" to "1" and "2", so you need to divide "ours = 1.03" by "1.2" and "1.59", then you get % ratio, and the savings are 100%-"value you get", so at the end, you get values 35.2% and 14.16%, not the ones you mention!!! Please revise line 358 accordingly, seems there is some problem also...
Formula (r ratio) in line 262. must be revised. No numbering. Also, efficiency is given as (?) normally (in %), but "r" also can be used. Describe the formula correctly - E - if this is energy, then it is not a "brightness", as it has a different measurement technique, so add also dimensions (kWh, Wh) or (cd/m2 or lux), etc. Give also a formula for "k", to be more specific.
Another issue is terminology, "streetlamp luminaire" - change to "street luminaire" everywhere. The lamp and luminaire for LED are the same. A lamp is the sodium vapour bulb for HPS luminaires, for LED we call it a light source, as it is LED PCB, not a lamp or bulb.
-For further research - it would be interesting to calculate/measure how much energy you consume, running this algorithm+training on the PC? Such self-consumption also must be addressed when compared to existing tech, as you have an external computing system that is not taken into account by energy consumption, so it is not a fair comparison. Please add also some approximate assumptions in the Discussion part. I believe it is worth investigating. What happens at the street or district level?
Author Response
Dear reviewer:
Many thanks to your comments and suggestions. Sorry for not stating that the "red part" of the revised manuscript is revised or deleted content, and the "blue part" is our new content, which may make it difficult for you to read. This time we have highlighted the revised content, hoping to make it easier for you to read. Blow are our responses to your comments
Point 1: Table 4, must be revised: "Energy saving efficiency" is wrongly calculated, as you now have some "energy consumption difference" not energy efficiency. The correct way is to compare to existing tech, respectively "ours" to "1" and "2", so you need to divide "ours = 1.03" by "1.2" and "1.59", then you get % ratio, and the savings are 100%-"value you get", so at the end, you get values 35.2% and 14.16%, not the ones you mention!!! Please revise line 358 accordingly, seems there is some problem also...
Response 1: Sorry for the mistake, this calculation method refers to an article that also studies road lighting system. We also realized that this calculation method is not correct. because our energy-saving efficiency seems to be too high. We have revised the relevant content to obtain the correct energy saving efficiency.
Point 2: Formula (r ratio) in line 262. must be revised. No numbering. Also, efficiency is given as (?) normally (in %), but "r" also can be used. Describe the formula correctly - E - if this is energy, then it is not a "brightness", as it has a different measurement technique, so add also dimensions (kWh, Wh) or (cd/m2 or lux), etc. Give also a formula for "k", to be more specific.
Response 2: After reading your last comments, we have removed the formula (r ratio), because it is only a simple estimate, and it is not accurate to compare the energy consumption of different systems in this way. Sorry for not explaining to you that the "red part" is the deleted or revised part, and the "blue part" is new content. We have submitted a highlighted version of the manuscript in hopes of making it easier to read.
Point 3: Another issue is terminology, "streetlamp luminaire" - change to "street luminaire" everywhere. The lamp and luminaire for LED are the same. A lamp is the sodium vapour bulb for HPS luminaires, for LED we call it a light source, as it is LED PCB, not a lamp or bulb.
Response 3: Thank you for your professional advice, we have carefully checked the article and revised the terminology.
Point 4: -For further research - it would be interesting to calculate/measure how much energy you consume, running this algorithm+training on the PC? Such self-consumption also must be addressed when compared to existing tech, as you have an external computing system that is not taken into account by energy consumption, so it is not a fair comparison. Please add also some approximate assumptions in the Discussion part. I believe it is worth investigating. What happens at the street or district level?
Response 4:
We have also considered these external influencing factors. The system will process images on the cloud server, and the training of the model is just a process of verifying the superiority of the model. The detection model stored on the cloud server has been trained with a large amount of data, and it can process images quickly. We believe his energy consumption is very small, which may be a problem for large street lighting systems. Similarly, the communication of street lamps also consumes some energy, but these energy consumptions are difficult to measure for a single system. For the large road lighting system, our idea is that it is not necessary for each street lamp to do image processing, but to share information by establishing communication among street lamps. For example, for street lamps on the same street, if a system detects traffic, it can send a signal to adjacent street lamps, allowing them to turn on the corresponding lighting mode in advance. This is also one of the research works we are going to do, and we have supplemented the relevant content in the article.
Best wishes,
Ren Tang
Reviewer 2 Report
The article is much better, but the question still arises as to why the power of the current system in Figure 14 fluctuates from day to day and is not constant. An appropriate explanation or indication of what kind of lighting system is currently in use would be necessary.
Author Response
Dear reviewer:
Many thanks to your comments and suggestions. Sorry for not stating that the "red part" of the revised manuscript is revised or deleted content, and the "blue part" is our new content, which may make it difficult for you to read. This time we have highlighted the revised content, hoping to make it easier for you to read. Blow are our responses to your comments.
Point 1: The article is much better, but the question still arises as to why the power of the current system in Figure 14 fluctuates from day to day and is not constant. An appropriate explanation or indication of what kind of lighting system is currently in use would be necessary.
Response 1: We use the electricity consumed by luminaires to represent its energy consumption. The overall power consumption of the system may be affected by some external factors, resulting in some fluctuations in energy consumption. These external factors are very complicated. For example, the temperature will affect the heat dissipation of the luminaires. Rainfall and windy weather will also bring some fluctuations in power consumption to the system, but the impact of these factors is very small. We have supplemented the relevant instructions in the article.
Best wishes,
Ren Tang